# Typical Guidelines for Well-Balanced Diet and Science Communication in Japan and Worldwide

**DOI:** 10.3390/nu16132112

**Published:** 2024-07-02

**Authors:** Naohisa Shobako, Hiroshi Itoh, Keiko Honda

**Affiliations:** 1Division of Food Science and Biotechnology, Graduate School of Agriculture, Kyoto University, Uji 611-0011, Kyoto, Japan; 2The Center for Preventive Medicine, Keio University, Minato-ku, Tokyo 106-0041, Japan; hiito@keio.jp; 3Laboratory of Medicine Nutrition, Kagawa Nutrition University, Sakado-City 350-0214, Saitama, Japan; honda@eiyo.ac.jp

**Keywords:** food guideline, dietary reference intake, traditional diet, science communication

## Abstract

Numerous studies have investigated healthy diets and nutrients. Governments and scientists have communicated their findings to the public in an easy-to-understand manner, which has played a critical role in achieving citizens’ well-being. Some countries have published dietary reference intakes (DRIs), whereas some academic organizations have provided scientific evidence on dietary methods, such as traditional diets. Recently, more user-friendly methods have been introduced; the Health Star Rating system and Optimized Nutri-Dense Meals are examples from Australia and Japan, respectively. Both organizations adopt a novel approach that incorporates nudges. This review summarizes the science communication regarding food policies, guidelines, and novel methods in Japan and other countries. In the food policies section, we discuss the advantages and disadvantages of the DRIs and food-based guidelines published by the government. Dietary methods widely known, such as The Mediterranean diet, Nordic diet, Japanese traditional diet, and the EAT-Lancet guidelines, were also reviewed. Finally, we discussed future methods of science communications, such as nudge.

## 1. Introduction

The remarkable progress of nutrition science has contributed to the considerable information on diets and ingredients that contribute to improved health. For more than 1000 years, since Hippocrates said, “Let food be thy medicine and medicine be thy food,” vitamins and other nutrients have been identified. Over the next 100 years, continued discoveries across a wide range of nutritional sciences, including molecular nutrition and epidemiology, led to the development of functional foods that might prevent several diseases. To transfer these findings, food policy and science communication play a critical role. Both the government and some academic societies have taken the initiative. For example, in Japan, the government has published two sets of dietary guidelines [1,2], and the Japan Atherosclerosis Society has also published *The Japan Diet* guidelines [3]. Science communication through guidelines is also active in developed countries, such as the U.S. and those in Europe. However, the high prevalence of lifestyle-related diseases remains a significant issue in these countries [4], including Japan [5,6]. Malnutrition is also becoming a major concern in Japan. Iizuka reported an excessive desire to be thin, to achieve a so-called “Cinderella weight” in Japan [7]. The infodemic is exacerbating this issue; social media influencers have been shown to significantly impact eating behavior and body image [8]. The Health and Nutrition Survey results reflect this phenomenon [9], introducing some shocking findings. Although many adults consume more than 8 g of salt daily, the most common response was, “I will not change my eating habits while having an interest.” The survey also highlights the lack of reduction in the prevalence of obesity and diabetes.

The purpose of this study was to review science communication to date, including guidelines, and to discuss what is needed to improve science communication.

## 2. Methods

We collected the dietary reference intakes (DRIs) and food-based guidelines from public agency websites. First, we reviewed guidelines from Japan, which represents the Asia region and has the longest life expectancy [10]. We also selected guidelines from countries we determined to be distinctive in North America, the Middle East, Europe, and Oceania. Additionally, we also searched for food-based guidelines whose effects were determined by human clinical trials. Finally, we reviewed two examples with novel approaches to science communication from recent articles.

## 3. Dietary Reference Intake

### 3.1. Overview

The government publishes a DRI for its citizens, which specifies the recommended daily amounts of nutrient consumption determined using several specialized indicators. The estimated average requirement (EAR) is the daily requirement that is estimated to be met by 50% of people in each segment. The recommended dietary allowance (RDA) is the amount of intake estimated to meet the daily needs of most individuals in each segment. For example, most of the RDAs were calculated using EAR + 2 standard deviations in the Japanese DRI [1]. Adequate intake (AI) is the quantity sufficient to maintain a good nutritional status for people in each segment when adequate scientific evidence is unavailable to calculate the EAR or RDA. The upper limit (UL) is the maximum amount of nutrient intake that would not induce overconsumption-associated health problems for most of the individuals in each segment. Some countries adopt estimated energy requirements (EERs) [11] or average requirements (Ars) in the energy section [12]. The indicators for which nutrients are recommended in the DRI vary country-wise. However, the spirit of evidence-based policy making (EBPM) can be seen in most countries’ DRIs. These guidelines are designed to cover a wide range of targets, with thresholds set for each sex and age group. Their common aim is to maintain the nation’s health and prevent lifestyle-related diseases, as described in the Japanese guideline [1].

Their values were designed through experts’ critical reviews. In the U.S., it has been reported that experts discussed the future possibility of incorporating systematic reviews and meta-analyses [13]. This process might contribute to creating more scientific and objective guidelines.

In this section, we first introduce a unique aspect of the Japanese DRI [1]. Next, we discuss DRIs from representative countries, such as the US (jointly with Canada) [14], the EU (EFSA DRI) [12], France [15], and Australia (jointly with New Zealand) [16]. The specific values for all nutrients are not listed because this was not the purpose of the present study.

### 3.2. DRIs from Japan and Other Countries

The Japanese DRI provides guidelines for recommended nutrient intake ranges for each age group [1] and is regularly updated by the Ministry of Health, Labor, and Welfare (MHLW) based on new studies. The Japanese DRI recommends the nutrient quantities required for diverse targets, including healthy people, and for preventing lifestyle-related diseases and frailty. The nutrients covered in the DRI include energy, protein, lipids (total fats, saturated fatty acids, n-6/n-3 lipids, etc.), carbohydrates, fiber, vitamins (A, D, E, K, B_1_, B_2_, niacin, B_6_, B_12_, folic acid, pantothenic acid, biotin, and C), and minerals (sodium, potassium, calcium, magnesium, phosphorus, iron, zinc, copper, manganese, iodine, selenium, chromium, and molybdenum). The notable points for each item are described below.

Sex-stratified estimated EERs have been established for three physical activity levels. Both excessive energy intake and energy scarcity are major health concerns in Japan. Cinderella weight has been described earlier, and the Japanese DRI mentioned the wide prevalence of underweight (BMI < 18.5 kg/m^2^), especially among young women.

Regarding fat intake, the Japanese DRI did not mention the recommended amounts of docosahexaenoic acid (DHA) or eicosapentaenoic acid (EPA), unlike other countries’ DRIs (Figure 1). Although the Japanese DRI references a representative meta-analysis of DHA and EPA, it was concluded that specifying a clear value was difficult, and the situation is similar for trans-fats (Figure 1). The Japanese DRI did not offer a recommendation regarding the LDL/HDL ratio, despite citing a well-known journal article about this topic [17].

In the lipid-soluble vitamin section, no standard has been established for choline in the Japanese DRI, in contrast to other countries. Although studies on the importance of choline have been published in Japan, they have not attracted much attention [18]. Efforts to avoid excessive vitamin D intake through establishing an AI that considers the vitamin D production in the skin by sunlight have been undertaken. The AI and UL were set for vitamin E, and the deficiency limits were not discussed, as a deficiency in this vitamin is not a major problem in Japan. Regarding vitamin K, the AI was the only specification. For water-soluble vitamins, the UL is not listed, although some studies on overdose have been published [19,20].

The salt intake range was typical in the micronutrient section. The latest version (2020 version) of the Japanese DRI sets a <7.5 and <6.5 g/day limit for adult men and women, respectively. The amount seems to be higher than other guidelines; however, separately strict ranges are provided for CKD and hypertension prevention.

For potassium, the Japanese DRI introduced the importance of consuming 3510 mg (90 mmol) per day based on the WHO guidelines [21] and a meta-analysis [22]. However, unique amounts, 2500 and 2000 mg AI for adult men and women, respectively, were listed because of the large discrepancy regarding the average intake of the Japanese people. For calcium, the EAR and RDA were calculated using a different method compared to that in the U.S., as the balance test is not conducted on Japanese people.

The Japanese DRIs had no large difference in trace metals, except for iodine, compared to those in other countries. A higher UL than that in the U.S. has been set because the Japanese consume more iodine from dietary sources, such as seaweed. Furthermore, this does not apply if people consume dietary items that are particularly high in iodine, such as kelp.

Although water is mentioned frequently in the DRI of each country, in Japan it is only introduced as a reference, and no specific target amount has been set. This is because, in Western countries, approximately 20–30% of water intake comes from food, whereas in Japan the ratio is approximately 50% of that intake [23], reflecting the unique Japanese culture. Another reason might be easy access to safe drinking water and the fact that the Japanese people drink a lot of tap water [24].

Next, we discuss DRIs outside of Japan. Differences between guidelines are probably most noticeable for lipids. In the U.S., standards are defined not only for n-3 and n-6 fatty acids but also for linoleic acid and α-linolenic acid [25]. The levels of cholesterol and trans fats have also been reported to be as low as possible. In Australia and New Zealand, AI of total fat, n-6 unsaturated fatty acids, and n-3 unsaturated fatty acids is set for infants (0–1 years). AI for other sections is specified for linoleic acid, α-linolenic acid, and total n-3 (DHA + EPA + docosapentaenoic acid [DPA]) [16]. In the protein section, recommendations for essential amino acids are also described in the U.S. [25] and France [26]. Regarding sugar, some DRIs outside of Japan discuss the amount: the U.S. [27], UK [28], and France [26]. However, some differences exist, such as limiting the definition to added sugars (the U.S.) and expanding it to free sugars (the UK).

### 3.3. Brief Summary

In this section, we discuss the major DRIs. Despite slight differences in the nutrients mentioned in each country’s DRI, all DRIs had one common characteristic: they all present clear values from previous research on the required dietary nutrient quantity for citizens. This clarity is advantageous for the DRIs in each country, albeit with the disadvantage of being difficult to utilize in real life. People must calculate the nutrient content of ingredients every time before cooking. Furthermore, not all nutrient content is listed on commercially available foods. Therefore, it is unrealistic to determine all the nutrient proportions that are consumed daily. This may be a reason for the discrepancy between ideal and actual nutrition intake. Table 1 summarizes each country’s DRI, as previously reviewed by Kishida [29].

## 4. Food-Based Guidelines

### 4.1. Overview

The standards defined for nutrient quantities are scientifically clear but difficult to incorporate into real-life applications. Therefore, food balance guides that recommend food ingredients have been adopted in many countries. In addition, various diets have been reported, and their guidelines were published.

### 4.2. Food Balance Guides from Japan and Other Countries

The Ministry of Agriculture, Forestry, and Fisheries (MAFF) of Japan designed a food balance guide [2]. Through easy-to-understand visual diagrams (Figure 2a), the guide recommends the daily consumption of vegetables, meat, fish, dairy products, and fruits. Unlike the DRI, this provides the advantage of visual representation to ascertain whether the meals are well balanced, although a disadvantage is that it is difficult to calculate how many servings of each menu item are used. Although some examples are provided at the end of the balance guide, the variation is too small to be considered in daily life.

Another unique point is that water and snacks are not mentioned, which should be considered separately in this balance guide. Therefore, it is difficult for untrained individuals to use a balance guide correctly.

Guidelines defined by foods rather than nutrients are utilized not only in Japan but also in many other countries.

In the U.S., the *My Plate Plan* has been proposed by the FDA and USDA [30]. The concept of visualizing the grains, vegetables, proteins, fruits, and dairy products that should be consumed along with the calories that are to be consumed is the same as that used in *The Spinning Top* in Japan (Figure 2b). However, the recommended doses vary slightly (Table 2). Additionally, the contents of added sugar, saturated fat, and sodium are included in the guidelines. Furthermore, the FDA and the USDA dietary guidelines for Americans [27] have features that are not found in Japanese guidelines, such as setting recommended amounts of vegetables by color and mentioning whole grains. The upper limit of sodium intake is 2300 mg (equivalent to 5.8 g of salt), which is lower than that in the Japanese DRI.

In the UK, the NHS proposed the *Eat Well Guide* (Figure 2c) [32], which comprises five sections: “Fruit and vegetables”, “Beans, pulses, fish, egg, meat, and other proteins”, “Potatoes, bread, rice, pasta, and other starchy carbohydrates”, “Dairy and alternatives”, and “Oil and spreads”. In particular, the fruit and vegetable section, which recommends five portions per day, is well-as the “5 a day guideline”. The principle underlying the UK’s guideline can be described as not requiring to be achieved during every meal. Achieving a good balance within a day or week is important. Therefore, the daily standard values are not stringently defined, except in the “Fruit and vegetable” section. In the protein section, two portions of fish, one of which is oily fish, per week, and consuming as little red meat as possible are recommended. In the carbohydrate section, the benefits of whole grains are explained. Moreover, the guideline clarifies that carbohydrates have fewer calories than fats and includes a consideration of excessive concerns about weight gain. Independent of the five main sections, the guideline recommends drinking plenty of fluids (6 to 8 cups) daily. This unique point differs from the Japanese and U.S. guidelines mentioned above. Additionally, fruit juice is recommended not to exceed 150 mL/day.

Similar food base guides have been issued in Middle Eastern countries. However, perhaps because of their unique food culture, their contents are characteristic. In Saudi Arabia, the *Healthy Food Palm* was issued by the SFDA (Figure 2d) [33]. As shown in Table 2, the upper limit of some foods, such as fruits and grains, was slightly higher than that of others. This may reflect the unique Arabian food culture, consuming diverse grains in one meal and with a preference for dates. It should also be noted that the level of academic reference is also, with many academic papers cited and clearly stated in the guidelines. However, Halawani reported that adherence to the guidelines was extremely low (26%) and should be improved [34].

In Australia, the NHMRC established the *Australian Guide for Healthy Eating* (Figure 2e) [35]. This was similar to that of the Japanese Food Guide Spinning Top, but the recommended amount differed slightly (Table 2). One of the unique features of this guideline is the recommendation of one or two meat-free meals per week.

As described above, worldwide, countries issue food-based guidelines. Only a few studies have demonstrated the effectiveness of this type of guideline. McCarthy reported that the U.S. *My Plate* guideline could ensure satiety for continuity, compared with calorie count [36]. Although significant changes in waist circumference were shown in some subgroups, continuous improvement in blood pressure was not observed. Fuller demonstrated that education based on the *Australian Guide to Healthy Eating* for 6 weeks successfully reduced body weight (2 kg) [37]. However, few research results are available to the best of our knowledge, and this might be one of the weak points of food-based guidelines.

Unlike the DRI, the advantage of this type of guideline is that it is easy to calculate from when cooking independently. On the other hand, however, the need to manage all cooking processes is a weak point of this guideline. It is unrealistic to live without any processed food or eating out; therefore, more realistic methods are required.

### 4.3. DASH Diet

The National Heart, Lung, and Blood Institute (NHLBI), a part of the NIH, suggests that the *Dietary Approach to Stop Hypertension* (DASH) diet contributes to the prevention and treatment of hypertension. As the name suggests, this diet is designed to manage blood pressure at appropriate levels. Similar to the food-based guidelines for each country presented in the previous section, the DASH diet recommends the amount of food that should be consumed in a day (or, in some cases, a week). However, numerous studies have evaluated the effectiveness of the DASH diet, which is supported by strong evidence. In 2020, Filippou reviewed 30 RCTs that evaluated the effect of the DASH diet on blood pressure, and their meta-analysis showed a significantly positive effect [38]. If limited to hypertensive persons, the significance remains unchanged. Based on a meta-analysis of cohort studies, Theodoridis reported that adherence to the DASH diet was associated with a reduction in blood pressure [39]. A large, long-term (median duration: 22 years) cohort study showed that adherence to the DASH diet reduced the risk of heart failure (HF) [40].

As shown in Table 2, the DASH diet has good nutritional balance, which may improve lipid metabolism, obesity, and blood pressure. Numerous reports of the various benefits of the DASH diet have been published. In 2021, Lari reviewed 54 clinical trials and concluded that the effects of the DASH diet included a reduction in body weight, BMI, waist circumference (WC), and total/LDL cholesterol [41]. Furthermore, Soltani’s meta-analysis supported the robust effects of BMI and WC [42]. A large-scale cohort study showed that adherence to the DASH diet was associated with a reduced risk of frailty in women [43]. Surprisingly, positive effects for neuro-psychological functions, such as sleep initiation, were reported by young women [44].

As shown in this section, the DASH diet has good potential not only for blood pressure management but also for regulating various other factors. Maintaining a high level of compliance is important to maximize these benefits. Kwan concluded that effective approaches beyond counseling alone should be investigated [45]. For example, Japanese medical researchers and a food company modified the limitations to customize the DASH diet for the Japanese culture (e.g., dietary salt was permitted for 8 g/day), and they succeeded in maintaining high compliance (88.5%) for 2 months [46]. Thus, the effects of the modified DASH diet were assessed in a single-arm study; however, more detailed studies are required, as well as activities to increase public awareness.

### 4.4. The Mediterranean Diet

The Mediterranean diet (MD) is a traditional meal consumed in Mediterranean countries, such as France, Italy, and Greece. Although the details vary slightly depending on the article and organization, the basic concept of the *Mediterranean Pyramid* comprises a detailed balance of food ingredients (Table 2). Although the high consumption of olive oil is well known, other characteristics include high fruit intake and a strict intake of red and processed meats (Figure 2f).

The MD is a traditional dietary pattern that has been evaluated for its health benefits. The historical origin of the research for this diet extends to *The Seven Countries Study* in the 1950s, which reported a remarkably low prevalence of coronary heart disease (CHD) [47]. Showing a robust weight-loss effect with a 2-year intervention, the comparison of low-fat and low-carbohydrate diets was an epoch-making discovery [48]. Despite a tendency to focus attention only on the anti-obesity effect in this study, it should not be overlooked that the MD showed the most favorable results in the sub-analysis of the diabetic group. Furthermore, another RCT showed the effect of diet on blood pressure and glucose metabolism compared with a low-fat diet [49]. *The Nu-Age trial*, a large-scale intervention study for evaluating the effects of frailty, is also well-known [50]. However, this clarified that a high compliance rate was also important.

Its potential is also well-studied compared with other diet theories. Lista indicated a cardiovascular disease prevention effect compared with the low-fat diet [51]. The PREDIMED trial, another comparison with a low-fat diet, reported multiple benefits of the MD, such as on cognitive function [52] and diabetes prevention [53].

Several organizations have focused on activities to promote the MD. The American Heart Association recommends the MD and provides some recipes on its homepage. The International Foundation of the Mediterranean Diet has focused on a more active awareness campaign and on publishing some academic papers [54,55].

### 4.5. Nordic Diet

In addition to the MD, Nordic diet (ND) has recently attracted considerable attention. It shares the same basic philosophy of consuming less processed red meat and more plant-based foods [56]. The most notable differences from the MD are: (1) the active use of canola oil instead of olive oil, (2) the intake of berries, and (3) the active intake of fatty fish, such as salmon.

The beneficial effects of ND have also been well documented. Kanerva et al. showed that high adherence to the ND was associated with weight change in a large-scale study in Finland [57]. In an RCT, gene expression associated with inflammation, such as LILRB2 and IL32, was found to be downregulated by the ND [58]. Another RCT showed significant weight and blood pressure reduction in obese participants compared with the average Danish diet [59]. The health benefits of ND are not limited to those related to obesity. The effects on frailty, particularly on muscle strength, have also been observed in older women [60]. Moreover, its effect on children has been well studied. Sørensen reported that 3 months of consumption of the ND improved school performance for children in a crossover interventional study [61]. Sabet reported improvement in depressive symptoms with an 8-day intervention for ND in an RCT [62].

Although ND has been reported to have various advantages, they are relatively new, as shown by the paucity of literature. Compared with the MD, there are issues regarding their recognition by the public. The Nordic Council of Ministers published the *Nordic Nutrition Recommendations* and renewed them in 2023 [63,64]. The basic principles of ND are introduced in this guideline. The issue is how far it will spread outside Nordic countries in the future.

### 4.6. Japanese Traditional Diet (Washoku)

*The Seven Countries Study* showed a low prevalence of CHD in Japan [47], and the traditional Japanese diet (washoku) has received attention worldwide. Unlike the MD and ND, there is no strict definition of washoku; however, there is a basic form that has reached a certain degree of consensus: one miso-based soup and three dishes with rice as the staple food [65]. It should be noted that dishes that are popular and special overseas, such as sushi, tempura, and donburi, are not included in washoku here. It is well known that Japanese dishes tend to contain a high salt content, which is thought to be one reason for the high prevalence of hypertension [66]. To solve this problem, several methods have been developed to retain the advantages of washoku, while incorporating improvements.

A well-known RCT of washoku was reported by Maruyama; serum total/LDL cholesterol and triglycerides were significantly reduced for 6 months compared to the partial intervention group, due to nutritional education about their washoku-based meals, named the *Japan diet* [3]. Moreover, Sakane reported beneficial effects for serum cholesterol and triglyceride metabolism of *Smart washoku*, their newly modified washoku, based on a crossover interventional study [67]. Animal studies conducted to clarify the mechanisms and changes in the expression of genes related to lipid metabolism, such as *Ucp2*, *Fabp4*, and *Fabp5*, were considered for association [68]. As shown in the papers cited in this chapter, there are few intervention studies on washoku, whereas several cohort studies other than *the Seven countries study* have been reported. For example, the Kashiwa study revealed a relationship between washoku and sarcopenia [69]. The JPHC study showed that washoku, containing a wide variety of ingredients, was associated with total mortality [70].

MAFF also promotes washoku; however, its main objective is to promote food culture rather than nutrition education. The Japan Atherosclerosis Society is promoting the *Japan diet* for humans with high LDL cholesterol or triglycerides; the balance for high LDL cholesterol is shown in Table 2. Other local Japanese governments and food companies are working to promote washoku, but the reality is that the Westernization of food has moved away from the traditional style of the 1970s, which was considered as having the best balance [71].

### 4.7. The EAT-Lancet Guideline

The EAT-Lancet guidelines (ELG) focus on health and sustainability [72]. Recently, prospective cohort studies evaluating health outcomes have been conducted. Langmann et al. reported that guideline adherence was associated with the prevalence of type 2 diabetes in the Danish population [73]. Zhang et al. reported similar results in a cohort study conducted in Sweden [74]. Tonstad et al. reported that, the higher the rate of plant-based protein sources, the lower the incidence of diabetes in a cohort study [75]. As shown in Table 2, this guideline recommends more plant-based and less animal meat as a protein source. Thus, a diet based on ELG might have a positive impact on diabetes, and the results of Langmann and Zhang are reasonable. Another cohort study indicated that meals based on ELG were not associated with cardiovascular disease, but reduced cancer risks in the female or low-alcohol consumption subgroups [76]. On the other hand, concerns remain that extreme restrictions on meat consumption may cause emotional stress [77].

Since this guideline focuses not only on wellness but also on sustainability, in just a few years it has attracted the interest of researchers in many fields, including sociologists and environmentalists, and its positive effects in their research area have been discussed [78]. However, some researchers have noted that citizens face challenges in incorporating these guidelines into their real lives. Adherence to this guideline is difficult for economically poor populations [79]. A solution to this issue is needed in the future.

### 4.8. Brief Summary

In this section, we discuss food-based guidelines. The advantages of these guidelines are that they are easy to reproduce because they are free from nutritional calculation. However, cultural differences and distances might pose challenges when obtaining foods described in these guidelines. For example, Tayyem compared the diet of Jordanian pregnant women with that recommended in the American MyPlate plan [80]. The majority of participants consumed more fruits and grains than the guideline recommendation. Notably, in the Saudi Arabia section, this divergence is understandable given the Middle Eastern culture, where dates and various staple foods are common. Another example is that in Japan, only 0.04% of total olive oil consumption is produced domestically [81]. To make matters worse, the price is increasing exponentially [82]. Therefore, it is economically difficult to consume olive oil in Japan, as recommended by the MD guidelines.

Thus, recommending food-based guidelines without considering foreign cultures and challenges is undesirable. A different approach with significantly improved science communication might be needed.

## 5. Recent Notable Consumer Communications

Although several systems have been reviewed, it is difficult to determine whether they provide sufficient information to consumers in real life. All of the methods described above require customer effort, such as calculating multiple factors. Efforts have been made to provide consumers with easy-to-understand visual information. Here, we introduce two typical examples of science communication.

The first is the Health Star Rating System (HSR) in Australia and New Zealand. This system reviews nutritional quality from a 0.5-point star (worst) to a 10-point star (best) [83]. In addition to the star points, the five columns indicate the energy, saturated fat, sugar, salt, and fiber contents. If each product meets certain criteria, it can be labeled “high” or “low”, making it easy for consumers to select “healthy products.” Jones reviewed government reports and journal articles and concluded that awareness was increasing, but campaign reach remained low, and the impact on consumer purchasing was unknown [84]. However, a large-scale cohort study showed that dietary patterns of primary foods with high HSR were associated with weight loss [85]. Pan et al. also demonstrated its positive effect on mortality in Australia [86]. The government led this system, and food companies voluntarily labeled the star points; however, Maganja revealed that only 14.3% of the products in online stores used by most of the public were labeled with the HSR [87]. He also pointed out that products with a higher HSR tend to display more star points and may be regarded by food companies as marketing tools. It is important to understand how governments can cooperate with food companies in the future.

The second is the Japan Optimized Nutri-Dense Meals Association (H. Itoh is the president). The association’s mission is to promote Optimized Nutri-Dense Meals and offer multiple choices of nutritionally well-balanced meals with good flavor, which are available at any time of the day. An Optimized Nutri-Dense Meal is defined as one in which major nutrients are balanced and appropriately adjusted for each target group of individuals, as specified by age, sex, lifestyle, and other criteria. For example, for obese people, meals with moderate or relatively low energy content, but adjusted to include sufficient nutrients with an appropriate balance of proteins, fats, and carbohydrates, should be served. For older people, a small meal, which could be consumed without difficulty, with an appropriate quantity of nutrients, such as proteins and calcium, and with sufficient energy content, should be provided. The association’s review is not limited to processed foods, but also applies to meal kits and menus served in restaurants.

More precisely, Optimized Nutri-Dense Meals should be designed such that, if one consumes only Optimized Nutri-Dense Meals all day and takes the energy required per day only from these meals, one can take adequate, for example, neither too much nor too little, amounts of the major nutrients required a day. Hence, one can take the proportional amount of the major nutrients according to the energy consumed from Optimized Nutri-Dense Meals, compared to the total daily energy required.

Thirty-three nutrients are required for approval (Table 3). The upper and lower limits of each nutrient were determined by sex, age, and characteristics of the target population. Each limit should be determined based on scientific evidence, such as research papers and DRIs for the Japanese population. Nutrients not listed in Table 3 are also permitted to be standardized. As of April 2024, two standards have been certified: one for age 18–64 and another for seniors. RCT design trials showed positive effects on hypertension, diabetes, and frailty prevention [88,89]. Multiple single-arm studies also demonstrated potential benefits, not only in physical parameters but also in work productivity [90].

Providers must ensure that, if the daily calorie requirement is to be obtained entirely from their meals, the amount of each nutrient should not exceed the upper and lower limits. Providers will also be required to verify the effects of their meals on health in human clinical trials, such as the study by Shobako et al. [88,89].

The certification committee performs meal certification audits to investigate whether the submitted meals satisfy registered nutritional standards. They allow approved meals to be labeled with an association-approval mark that customers can easily recognize (Figure 3). As this system has just been implemented, it is important to determine how far it can modulate people’s lives.

## 6. Conclusions and Further Perspective

Many countries have set DRIs for national health improvement. Such countries have also published a parallel set of food-based guidelines to make it easier for individuals to reproduce them when cooking. This was accompanied by visually clear reference diagrams of spins, palm trees, and circles. Unique dietary patterns, such as the MD, DASH diet, and washoku (Japanese diet), are also being devised by academic societies and associations for communication with consumers. Furthermore, simple systems such as HSR and approval marks for Optimized Nutri-Dense Meals are now available. We hope this initiative will spread worldwide soon. However, efforts to raise awareness of these factors are needed.

Therefore, governments and food companies need to communicate more closely with consumers. Science communication can be considered in multiple ways. Maruyama classified the methods of science communication into four types: information, financial, regulatory, and nudge [91]. The information approach is a classical style of providing information to the public. The DRIs and guidelines discussed in this review provide typical examples. Financial approaches have a long history in nutritional science communication; the soda tax is an epochal example. Several studies show that soda taxes effectively reduce sweet beverage consumption [92,93], but the opposite result has also been reported [94]. Wright reviewed that a tax burden of more than 20% is critical for the financial approach to be effective [95]. Momin pointed out that the balance between effectiveness and ethical aspects must be carefully judged, and there are hurdles to its implementation [96]. Furthermore, difficulties in maintaining support have been studied, depending on the timing of exposure of the oppositional message from industry [97]. A regulatory approach imposes strong public restrictions. The legal prohibition of trans fats in the U.S. exemplifies this. Partially hydrogenated oils, a major source of trans fats, are also prohibited in Thailand. Chavasit reported a significant reduction in trans fats in distributed bakery products [98]. However, it is impractical to apply this method to individual dietary patterns. In recent years, nudges have received considerable attention in the field of public health. Detailed experimentation is described in Maruyama’s article; simply, this is a “gentle poke” aimed at an individual or group’s decision-making, without coercion. The introduction of simple symbols is a well-known example of nudge. This visual effect is more effective in motivating people rather than authoritative commands; a social examination of social distancing during the COVID-19 pandemic era demonstrated this [99]. Ellis reported that Het Vinkje contributed to consumer’s choice [100]. Het Vinkje is a mark approved for foods recognized in the Netherlands as having a low number of unhealthy components, such as saturated fatty acids [101]. Of course, examples of nudges are not limited to this. A nudge can also provide information that reminds people of their “ownership”. This approach has been noted for promoting COVID-19 vaccination [102] wherein information emphasizing that “the vaccine was prepared for you” stimulated ownership and accelerated COVID-19 vaccine uptake. Considering these examples, HSR and Optimized Nutri-Dense Meals, and labeling with eye-catching symbols, might be a noteworthy example of nudge in nutritional science. Since the target was subdivided to easily realize the idea “this product is for me”, Optimized Nutri-Dense Meals might be a more in-depth nudge example. We hope that science communication in dietary methods will develop beyond a mere “information approach” in the future. For further study, we aim to investigate its contribution to behavioral changes and health status.

## 7. Limitation

Our review had two limitations. First, it was not a systematic review. Countries and guidelines were selected based on the author’s interest. For example, we did not discuss guidelines in African countries, as recently reviewed by Anuson-Quampah [103]. In particular, traditional diets in Nigeria were reviewed in depth [104]. However, major guidelines are included in this article. Second, we focused on DRIs only for healthy young to middle-aged people. For instance, the Japanese DRI has established detailed classification and standard values for people over 50 years. While these details are important, our focus was on comparing various communications, such as guidelines. We hope that our review will contribute to the development of science communication.

## Figures and Tables

**Figure 1 nutrients-16-02112-f001:**
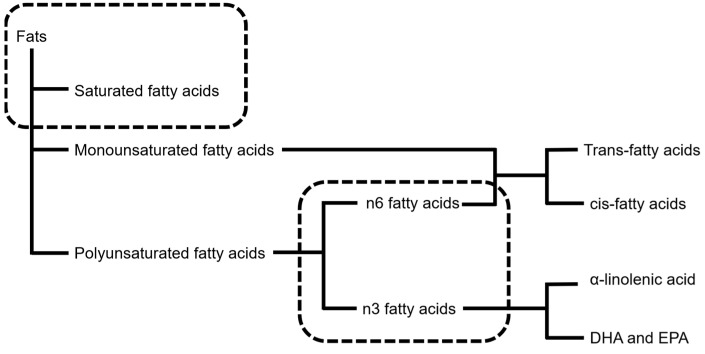
Fats (recommended amount) described in Japanese DRI. The fats enclosed by the dotted rectangle are regulated in the Japanese DRI.

**Figure 2 nutrients-16-02112-f002:**
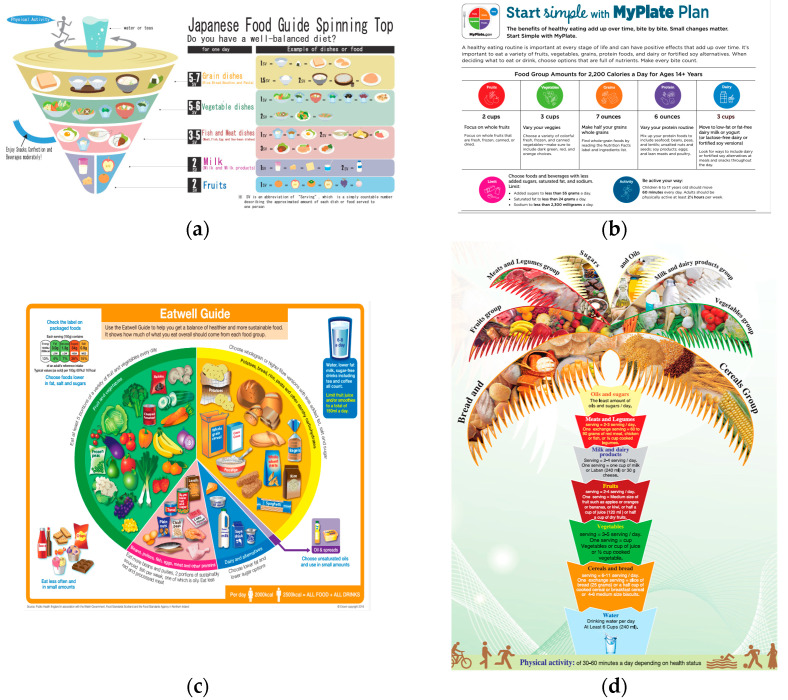
Food balance guides from various countries and traditional diets. (**a**) Japanese Food Guide Spinning Top. (**b**) My plate plan in the USA. (**c**) The Eat well guide in the UK. (**d**) The healthy food palm in Saudi Arabia. (**e**) The Australian guide to healthy eating. (**f**) The Mediterranean diet.

**Figure 3 nutrients-16-02112-f003:**
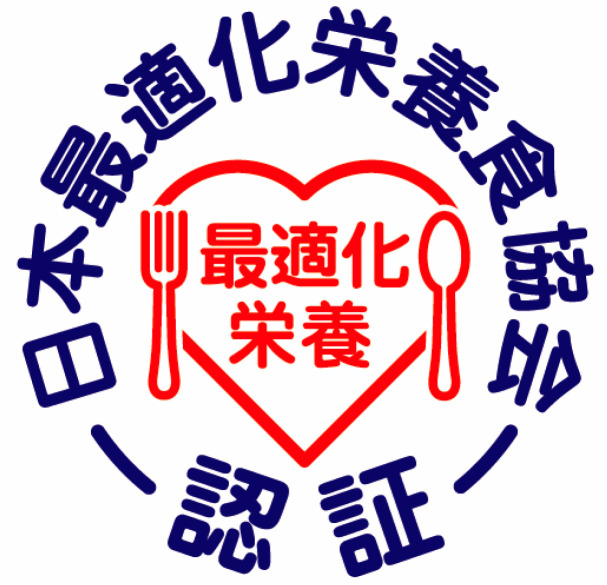
Approval symbol for Optimized Nutri-Dense Meals. Its definition is discussed in Section 5.

**Table 1 nutrients-16-02112-t001:** Differences in nutrients specified in major DRIs. The energy column shows the indicators considered for reference values. Other columns indicate whether the values were defined or not.

Nutrients		Japan	U.S.	EU	UK	France *	Australia/New Zealand
Energy		BMI	EER	AR	EAR	EER	EER
Protein		〇	〇 (+Amino Acids)	〇	〇	〇 (+Amino Acids)	〇
Fats	Total fats	〇	〇	〇	〇	〇	〇 (Infants only)
	SFA	〇	〇	〇 ****	〇	〇	× **
	Mono unsaturated fatty acids	×	×	×	〇	×	×
	Poly unsaturated fatty acids	×	×	×	〇	×	×
	n6	〇	〇	×	×	×	〇 (0–12 month)
	n3	〇	〇	×	×	×	〇 (0–12 month)
	Linoleic acid	×	〇	〇	〇	〇	〇 (age 1 or more)
	α-linolenic acid	×	〇	〇	〇	〇	〇 (age 1 or more)
	DHA, EPA	×	×	〇	×	〇	〇 ^†^
	Cholesterol	〇 ***	〇 ****	×	×	×	×
	Trans	×	〇 ****	〇 ****	〇	〇	× **
Carbohydrate	Carbohydrate	〇	〇	〇	〇	〇	〇 (Infants only)
	Fiber	〇	〇	〇	〇	〇	〇
	Sugars	×	〇 (Added sugars)	×	〇 (Free sugars)	〇 (Total sugars)	×
Vitamin	A	〇	〇	〇	〇	〇	〇
	D	〇	〇	〇	〇	〇	〇
	E	〇	〇	〇	〇	〇	〇
	K	〇	〇	〇	〇	〇	〇
	B_1_	〇	〇	〇	〇	〇	〇
	B_2_	〇	〇	〇	〇	〇	〇
	Nia	〇	〇	〇	〇	〇	〇
	B_6_	〇	〇	〇	〇	〇	〇
	B_12_	〇	〇	〇	〇	〇	〇
	FA	〇	〇	〇	〇	〇	〇
	PA	〇	〇	〇	〇	〇	〇
	Biotin	〇	〇	〇	〇	〇	〇
	Choline	×	〇	〇	×	〇	〇
	C	〇	〇	〇	〇	〇	〇
Minerals	Na	〇	〇	〇	〇	〇	〇
	K	〇	〇	〇	〇	〇	〇
	Ca	〇	〇	〇	〇	〇	〇
	Mg	〇	〇	〇	〇	〇	〇
	P	〇	〇	〇	〇	〇	〇
	Fe	〇	〇	〇	〇	〇	〇
	Zn	〇	〇	〇	〇	〇	〇
	Cu	〇	〇	〇	〇	〇	〇
	Mn	〇	〇	〇	〇	〇	〇
	I	〇	〇	〇	〇	〇	〇
	Se	〇	〇	〇	〇	〇	〇
	Cr	〇	〇	×	〇	〇	〇
	Mo	〇	〇	〇	〇	〇	〇
	F	×	〇	〇	〇	〇	〇
	B	×	〇	×	〇	×	×
	Ni	×	〇	×	×	〇	×
	V	×	〇	×	×	×	×
	Cl	×	〇	〇	〇	〇	×
	As	×	×	×	×	〇	×
Water		×	〇	〇	×	〇	〇

〇: Value was defined. ×: Value was not defined. * Some standards of nutrients were available on the ANSES homepage, but they were not English-translated guidelines (https://www.anses.fr/en/content/dietary-reference-values-vitamins-and-minerals, accessed on 26 June 2024). ** In the main text, a combined limit of 8–10% of energy from saturated and trans fats together is recommended. *** In the abstract, the specific standards were not set. However, the recommendation of less than 200 mg/day was described in the main text. **** As low as possible. ^†^ DHA + EPA + DPA. SFA, saturated fats; DHA, docosahexaenoic acid; DPA, docosapentaenoic acid; EPA, eicosapentaenoic acid; Nia, niacin; Na, sodium; K, potassium; Ca, calcium; Mg, magnesium; P, phosphorus; Fe, iron; Zn, zinc; Cu, copper; Mn, manganese; I, iodine; Se, selenium; Cr, chromium; Mo, molybdenum; F, fluorine; B, boron; Ni, nickel; V, vanadium; Cl, chlorine; As, arsenic.

**Table 2 nutrients-16-02112-t002:** Comparison of food-based guidelines. Recommended daily amount unless otherwise stated.

Food Guidelines		Japan	U.S.	UK	Saudi Arabia	Australia	DASH	The Mediterranean Diet **	The Japan Diet ***	The EAT-LANCET
Kcal		2200	2200						2200	2500
Units		serve	cups	portion	serve	serve	serve	serve	grams	grams
Grains		5–7	7 (ounce)		6–11	6	6–8			
	Whole grain		3.5 (ounce)					1–2/meal		232
	Refined grain		3.5 (ounce)					≤3/week	440 (mixed with rolled oats) 250 (if “soba” choosed)	
Vegetable		5–6	3	5	3–5	5–6	4–5	2-/meal	170 (green and yellow) 230 (others)	300 (200-600)
	Dark-green		2/week							
	Red and orange		6/week							
	Beans		2/week							
	Starchy		6/week						110	50 (0–100)
	Other		5/week						60 (seaweed and mushroom)	
Protein		3–5	6 (ounce)		2–3	2.5–3	6 or less			
	Meats, poultry, eggs		28 (ounce) /week					eggs: 2–4/week poultry: 1–2/week red meat: ≤2/week processed meat: ≤1/week	eggs: 10 meats (without chicken skin and fat): 80	meat: 43 (0–86) eggs: 13 (0–25)
	Seafood		9 (ounce) /week	2/week *				2-/week	100	28 (0–100)
	Nuts, seeds, soyproducts		5 (ounce) /week				4–5/week	legumes: 2-/week olive, nuts, seeds: 1–2	soy: 100	nuts: 125 (0–175)
Dairy products		2	3		2–4	2.5	2–3 (low fat)	2–3/meal (low fat preferably)	210	250 (0–500)
Fruits		2	2		2–4	2	4–5	1–2/meal	275	200 (100–300)
Oils			29 (grams)		the least amount				27	Unsaturated: 40 (20–80) Saturated: 11.8 (0–11.8)
Extra virgin olive oil								3–4		
Fluids				6–8	6 (cups)					
Sodium (mg)			2300				2300 /1500		22 (salt + miso + soysauce)	
Sweets					sugar: the least amount		5 or less/week	≤2/week	sugar:12	Sugar: 31 (0–31)

* One of which is oily. ** Varies slightly in the literature; however, we refer to the article by Alessandro et al. (2019) [31]. *** We refer to the “for high LDL-cholesterol” version, whereas the “for high triglyceride” version differs slightly.

**Table 3 nutrients-16-02112-t003:** Nutrients that need to be standardized for Optimized Nutri-Dense Meals.

Nutrients	Lower Limit	Upper Limit	Nutrients	Lower Limit	Upper Limit
Protein			Niacin		
Fats			Vitamin B_6_		
*n*-3 fatty acid			Vitamin B_12_		
Carbohydrate			Folic acid		
Sodium			Pantothenic acid		
Potassium			Biotin		
Calcium			Vitamin C		
Magnesium			Isoleucine		
Iron			Leucine		
Zinc			Lysine		
Copper			Sulfur-containing amino acid		
Vitamin A			Aromatic amino acid		
Vitamin D			Threonine		
Vitamin E			Tryptophan		
Vitamin K			Valine		
Vitamin B_1_			Histidine		
Vitamin B_2_

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
