# Peer review of "Typical Guidelines for Well-Balanced Diet and Science Communication in Japan and Worldwide"

_nutrients, 2024, doi:10.3390/nu16132112_

Round 1
Reviewer 1 Report
Comments and Suggestions for Authors
1. Introduction: It is recommended that the authors include detailed information on the characteristics of the Japanese Dietary Guidelines and articulate the research significance of this study. Is there a necessity for improvements in the current overall nutritional intake in Japan? Alternatively, should the focus of this research be directed towards specific areas? Additionally, are there existing research gaps concerning Japan’s recommended dietary guidelines that this study addresses?
2. Literature Review: The manuscript lacks a comprehensive literature review that underpins the development of a theoretical framework for dietary guidelines.
3. Comparative Analysis: The article mentions dietary guidelines from several countries, such as Australia and Japan. It would be beneficial for the authors to delineate the similarities and differences between these guidelines. A comparative table should be created for a clear analysis, and the rationale for selecting these particular countries as representatives should be thoroughly explained.
4. Methodology: The methodology section is particularly weak and requires substantial improvement. The authors need to provide a more detailed and rigorous explanation of the methodological approach. This should include a comprehensive description of the study design, data collection methods, and analytical techniques employed. Strengthening this section is crucial for the validity and reliability of the study's findings.
5. Discussion: The discussion section would benefit from incorporating a comparative analysis with previous literature. This would provide a broader context and enhance the robustness of the findings.
6. Research Limitations: The manuscript does not adequately address the limitations of the study. A section discussing potential limitations and their implications is essential.
7. Contributions and Recommendations: It is recommended that the authors elaborate on the contributions of the study and provide practical recommendations for nutritional intake practices based on their findings.
Author Response
To reviewer 1
Thank you for reviewing our manuscript. All changes are indicated in red.
>> Comment 1. Introduction: It is recommended that the authors include detailed information on the characteristics of the Japanese Dietary Guidelines and articulate the research significance of this study. Is there a necessity for improvements in the current overall nutritional intake in Japan? Alternatively, should the focus of this research be directed towards specific areas? Additionally, are there existing research gaps concerning Japan’s recommended dietary guidelines that this study addresses?
According to your suggestion, we have rewritten and modified the introduction to indicate the current issues about nutrition intake and DRIs in Japan. We also added references supporting this.
Line 34-52: To transfer these findings, food policy and science communication play a critical role. Both the government and some academic societies have taken the initiative. For example, in Japan, the government has published two sets of dietary guidelines [1,2], and the Japan Atherosclerosis Society has also published “The Japan Diet” guidelines[3]. Science communication through guidelines is also active in developed countries such as the U.S. and those in Europe. However, the high prevalence of lifestyle-related diseases remains a significant issue in these countries[4], including Ja-pan[5,6]. Malnutrition is also becoming a major concern in Japan. Iizuka reported an excessive desire to be thin, a so-called “Cinderella weight” in Japan[7]. The infodemic is exacerbating this issue; social media influencers have been shown to significantly impact eating behavior and body image[8]. The Health and Nutrition Survey results reflect this phenomenon [9], introducing some shocking findings. Although many adults consume more than 8 g of salt daily, the most common response was, “I will not change my eating habits while having an interest.” The survey also highlights the lack of reduction in the prevalence of obesity and diabetes.
The purpose of this study was to review science communication to date, including guidelines, and to discuss what is needed to improve science communication.
>>Comment 2. Literature Review: The manuscript lacks a comprehensive literature review that underpins the development of a theoretical framework for dietary guidelines.
According to your suggestion, we added a detailed review of the concept of DRIs not only in Japan.
Line 75-85: Some countries adopt estimated energy requirements (EER)[11] or average requirements (ARs) in the energy section[12]. The indicators for which nutrients are recommended in the DRI vary country-wise. However, the spirit of evidence-based policy making (EBPM) can be seen in most countries' DRIs. These guidelines are designed to cover a wide range of targets, with thresholds set for each sex and age group. Their common aim is to maintain the nation’s health and prevent lifestyle-related diseases, as described in the Japanese guideline[1].
Their values were designed through experts’ critical reviews. In the U.S., it has been reported that experts discussed the future possibility of incorporating systematic reviews and meta-analyses[13]. This process might contribute to creating more scientific and objective guidelines.
>>Comment 3 Comparative Analysis: The article mentions dietary guidelines from several countries, such as Australia and Japan. It would be beneficial for the authors to delineate the similarities and differences between these guidelines. A comparative table should be created for a clear analysis, and the rationale for selecting these particular countries as representatives should be thoroughly explained.
According to your suggestion, we created a novel table comparing the DRIs. Please check “table1” in our revised manuscript. The selection method and its biases are also added in the novel sections “methods” and “limitation”; reply to comments 4 and 6, respectively.
In addition, we added the notable points revealed by the comparison.
Line 150-161: Next, we discuss DRIs outside of Japan. Differences between guidelines are probably most noticeable in lipids. In the U.S., standards are defined not only for n-3 and n-6 fatty acids but also for linoleic acid and α-linolenic acid[26]. The levels of cholesterol and trans fats have also been re-ported to be as low as possible. In Australia and New Zealand, AI of total fat, n-6 unsaturated acids, and n-3 unsaturated acids is set for infants (0–1 years). AI for oth-er sections is specified for linoleic acid, α-linolenic acid, and total n-3 (DHA+EPA+ docosapentaenoic acid [DPA])[27]. In the protein section, recommendations for es-sential amino acids are also described in the U.S.[26] and France[28]. Regarding sug-ar, some DRIs outside of Japan discuss the amount: the U.S.[29]., U.K.[30], and France[28]. However, some differences exist, such as limiting the definition to added sugars (the U.S.) and expanding it to free sugars (the U.K.).
>>Comment 4. Methodology: The methodology section is particularly weak and requires substantial improvement. The authors need to provide a more detailed and rigorous explanation of the methodological approach. This should include a comprehensive description of the study design, data collection methods, and analytical techniques employed. Strengthening this section is crucial for the validity and reliability of the study's findings.
According to your suggestion, we added the “method” section below.
Line 53-61: 2. Methods
We collected the dietary reference intakes (DRIs) and food-based guidelines from public agency websites. First, we reviewed guidelines from Japan, which rep-resents the Asia region and has the longest life expectancy[10]. We also selected guidelines from countries we determined to be distinctive in North America, the Middle East, Europe, and Oceania. Additionally, we also searched for food-based guidelines whose effects were determined by human clinical trials. Finally, we re-viewed two examples with novel approaches to science communication from recent articles.
As described, our manuscript was not a systematic review but just a narrative review. Thus, we explained its basis in the limitation section. This also might be one answer for comment 6.
>>Comment 5. Discussion: The discussion section would benefit from incorporating a comparative analysis with previous literature. This would provide a broader context and enhance the robustness of the findings.
According to your suggestion, we added more references in the discussion section. We would like to thank you for emphasizing the importance of nudges more by this revision.
Line 518-521: Several studies show that soda taxes effectively reduce sweet beverage consumption[94,95], but the opposite result has also been reported[96]. Wright reviewed that a tax burden of more than 20% is critical for the financial approach to be effective[97].
Line 523-525: Furthermore, difficulties in maintaining support have been studied, depending on the timing of exposure of the opposition message from industries[99].
Line 527- 529: Partially hydrogenated oils, a major source of trans fats, are also prohibited in Thailand. Chavasit reported a significant reduction in trans fats in distributed bakery products[100].
Line 534-536: This visual effect is more effective in motivating people rather than authority; a social examination of social distancing during the COVID-19 pandemic era demonstrated this[101].
>>Comment 6. Research Limitations: The manuscript does not adequately address the limitations of the study. A section discussing potential limitations and their implications is essential.
According to your suggestion, we added “limitation” section.
Line 551-561: 7. Limitation
Our review had two limitations. First, it was not a systematic review. Countries and guidelines were selected based on the author’s interest. For example, we did not dis-cuss guidelines in African countries, as recently reviewed by Anuson-Quampah[105]. In particular, traditional diets in Nigeria were deeply reviewed[106]. However, major guidelines are likely included in this article. Second, we focused on DRIs only for healthy young to middle-aged people. For instance, the Japanese DRI has established a detailed classification and standard values for people over 50 years. While these details are important, our focus was on comparing various communications, such as guidelines. We hope that our review will contribute to the development of science communication.
>>Comment 7. Contributions and Recommendations: It is recommended that the authors elaborate on the contributions of the study and provide practical recommendations for nutritional intake practices based on their findings.
We presented the potential for nudges to solve science communication problems related to nutrition in the future. According to your comment, we added further studies about Nudge introducing communication; the Optimized Nuri-Dense Meals
Line 548-550: For further study, we would like to investigate its contribution to behavioral changes and health status.
Sincerely,

Reviewer 2 Report
Comments and Suggestions for Authors
Naohisa Shobako and colleagues present an interesting review about typical general guidelines for well-balanced diets.
This review highlights the critical role of scientific communication in promoting healthy dietary habits and well-being among citizens. By summarizing dietary guidelines and methods across different countries, the review provides a comprehensive overview in nutritional science communication.
This review could benefit if the scope and objectives of the guidelines of each country were previously stated.
It would also be useful to include a comparative analysis of the principal key differences and similarities between the different guidelines.
Also more detailed evidence and more examples from the various studies mentioned would strengthen the review.
Recognizing cultural and regional variations in dietary habits and how they influence the effectiveness of different dietary methods would improve the review.
Overall, this review contributes to the field of nutritional science communication bringing the current methods and future sugestions in promoting healthy diets, globally.
Author Response
To reviewer 2
Thank you for reviewing our manuscript. All changes are indicated in red.
>> This review could benefit if the scope and objectives of the guidelines of each country were previously stated.
According to your suggestion, we added the purpose of the guidelines described in them.
Line 76-81: The indicators for which nutrients are recommended in the DRI vary country-wise. However, the spirit of evidence-based policy making (EBPM) can be seen in most countries' DRIs. These guidelines are designed to cover a wide range of targets, with thresholds set for each sex and age group. Their common aim is to maintain the nation’s health and prevent lifestyle-related diseases, as described in the Japanese guideline[1].
>> It would also be useful to include a comparative analysis of the principal key differences and similarities between the different guidelines.
We created a novel table comparing the DRIs, according to your suggestion. Please check “table1” in our revised manuscript. (Comparing food-based guidelines was also described and moved to table2)
Additionally, we added notable points revealed by comparison.
Line: 150-161: Next, we discuss DRIs outside of Japan. Differences between guidelines are probably most noticeable in lipids. In the U.S., standards are defined not only for n-3 and n-6 fatty acids but also for linoleic acid and α-linolenic acid[26]. The levels of cholesterol and trans fats have also been reported to be as low as possible. In Australia and New Zealand, AI of total fat, n-6 unsaturated acids, and n-3 unsaturated acids is set for infants (0–1 years). AI for other sections is specified for linoleic acid, α-linolenic acid, and total n-3 (DHA+EPA+ docosapentaenoic acid [DPA])[27]. In the protein section, recommendations for essential amino acids are also described in the U.S.[26] and France[28]. Regarding sugar, some DRIs outside of Japan discuss the amount: the U.S.[29]., U.K.[30], and France[28]. However, some differences exist, such as limiting the definition to added sugars (the U.S.) and expanding it to free sugars (the U.K.).
>> Also more detailed evidence and more examples from the various studies mentioned would strengthen the review.
According to your comment, we added references, allowing us to deepen our discussion. (the number of references increased 75 to 106)
Detail references about the current issue of nutrition intake and DRIs.
Line 40-41: However, the high prevalence of lifestyle-related diseases remains a significant issue in these countries[4], including Japan[5,6].
Detail references about infodemics
Line 42-45: Iizuka reported an excessive desire to be thin, a so-called “Cinderella weight” in Japan[7]. The infodemic is exacerbating this issue; social media influencers have been shown to significantly impact eating behavior and body image[8].
Detail references about the water intake of the Japanese.
Line 145-149: This is because, in Western countries, approximately 20–30% of water intake comes from food, whereas in Japan, the ratio is approximately 50% of that intake [24]; reflecting the unique Japanese culture. Another reason might be easy access to safe drinking water and the fact that the Japanese people drink a lot of tap water[25].
Detail reference about the Mediterranean diet.
Line 336-339: Its potential is also well-studied compared with other diet theories. Lista indicated cardiovascular disease prevention effect compared with the low-fat diet[53]. PREDIMED trial, another comparison with a low-fat diet, reported multiple benefits of MD, such as cognitive function[54] and diabetes prevention[55].
Detail reference about the Optimized Nutri-Dense Meals.
Line 484-487: RCT design trials showed positive effects on hypertension, diabetes, and frailty prevention[90,91]. Multiple single-arm studies also demonstrated potential benefits, not only in physical parameters but also in work productivity[92].
Detail reference about the merits and demerits of soda tax.
Line 518-521: Several studies show that soda taxes effectively reduce sweet beverage consump-tion[94,95], but the opposite result has also been reported[96]. Wright reviewed that a tax burden of more than 20% is critical for the financial approach to be effective[97].
Line 523-525: Furthermore, difficulties in maintaining support have been studied, depending on the timing of exposure of the opposition message from industries[99].
Detail reference about the effect of the prohibition approach
Line 527-529: Partially hydrogenated oils, a major source of trans fats, are also prohibited in Thailand. Chavasit reported a significant reduction in trans fats in distributed bakery products[100].
Detail reference about Nudge in the COVID-19 pandemic era.
Line 534-536: This visual effect is more effective in motivating people rather than authority; a social examination of social distancing during the COVID-19 pandemic era demonstrated this[101].
>>Recognizing cultural and regional variations in dietary habits and how they influence the effectiveness of different dietary methods would improve the review.
This is the most critical matter to consider the science communication in nutrition science. We believe that we should change from the one-way "information approach" to the novel nudge challenge.
According to your suggestion, we added the discussion.
Line 422-436: In this section, we discuss food-based guidelines. The advantages of these guidelines are that they are easy to reproduce because they are free from nutrition calculation. However, cultural differences and distances might pose challenges when obtaining foods described in these guidelines. For example, Tayyem compared the diet of Jordanian pregnant women with that recommended in the American MyPlate plan[82]. The majority of participants consumed more fruits and grains than the guideline recommendation. Notably, in the Saudi Arabia section, this divergence is understandable given the Middle Eastern culture, where dates and various staple foods are common. Another example is that in Japan, only 0.04% of total olive oil consumption is produced domestically[83]. To make matters worse, the price is increasing exponentially[84]. Therefore, it is economically difficult to consume olive oil in Japan, as recommended by the MD guidelines.
Thus, recommending food-based guidelines without considering foreign cultures and challenges is undesirable. A different approach with noble science communication might be needed.
Line 548-550: For further study, we aim to investigate its contribution to behavioral changes and health status.
Sincerely,

Round 2
Reviewer 1 Report
Comments and Suggestions for Authors
Thanks to the author for the revision.